# Lifelong Learning with Dynamically Expandable Networks

**Jaehong Yoon**[1*]**, Eunho Yang**[1,3]**, Jeongtae Lee**[2]**, Sung Ju Hwang**[1,3]
KAIST[1], Daejeon, South Korea, UNIST[2], Ulsan, South Korea, AITrics[3], Seoul, South Korea
`jaehong.yoon93@gmail.com, jtlee@unist.ac.kr`
`{eunhoy, sjhwang82}@kaist.ac.kr`

## Abstract

We propose a novel deep network architecture for lifelong learning which we refer to as Dynamically Expandable Network (DEN), that can dynamically decide its network capacity as it trains on a sequence of tasks, to learn a compact overlapping knowledge sharing structure among tasks. DEN is efficiently trained in an online manner by performing selective retraining, dynamically expands network capacity upon arrival of each task with only the necessary number of units, and effectively prevents semantic drift by splitting/duplicating units and timestamping them. We validate DEN on multiple public datasets under lifelong learning scenarios, on which it not only significantly outperforms existing lifelong learning methods for deep networks, but also achieves the same level of performance as the batch counterparts with substantially fewer number of parameters. Further, the obtained network fine-tuned on all tasks obtained siginficantly better performance over the batch models, which shows that it can be used to estimate the optimal network structure even when all tasks are available in the first place.

## 1 Introduction

Lifelong learning (Thrun, 1995), the problem of continual learning where tasks arrive in sequence, is an important topic in transfer learning. The primary goal of lifelong learning is to leverage knowledge from earlier tasks for obtaining better performance, or faster convergence/training speed on models for later tasks. While there exist many different approaches to tackle this problem, we consider lifelong learning under deep learning to exploit the power of deep neural networks. Fortunately, for deep learning, storing and transferring knowledge can be done in a straightforward manner through the learned network weights. The learned weights can serve as the knowledge for the existing tasks, and the new task can leverage this by simply sharing these weights.

Therefore, we can consider lifelong learning simply as a special case of online or incremental learning, in case of deep neural networks. There are multiple ways to perform such incremental learning (Rusu et al., 2016; Zhou et al., 2012). The simplest way is to incrementally fine-tune the network to new tasks by continuing to train the network with new training data. However, such simple retraining of the network can degenerate the performance for both the new tasks and the old ones. If the new task is largely different from the older ones, such as in the case where previous tasks are classifying images of animals and the new task is to classify images of cars, then the features learned on the previous tasks may not be useful for the new one. At the same time, the retrained representations for the new task could adversely affect the old tasks, as they may have drifted from their original meanings and are no longer optimal for them. For example, the feature describing stripe pattern from *zebra*, may changes its meaning for the later classification task for classes such as *striped t-shirt* or *fence*, which can fit to the feature and drastically change its meaning.

Then how can we ensure that the knowledge sharing through the network is beneficial for all tasks, in the online/incremental learning of a deep neural network? Recent work suggests to either use a regularizer that prevents the parameters from drastic changes in their values yet still enables to find a good solution for the new task (Kirkpatrick et al., 2017), or block any changes to the old task

---

[*]work done while at UNIST

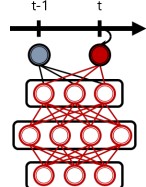 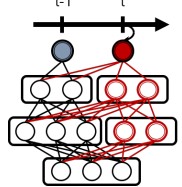 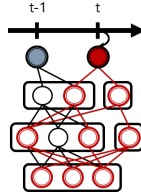

(a) Retraining w/o expansion  (b) No-retraining w/ expansion  (c) Partial retraining w/ expansion

Figure 1: **Concept:** (a) Retraining models such as Elastic Weight Consoliation Kirkpatrick et al. (2017) retrains the entire network learned on previous tasks while regularizing it to prevent large deviation from the original model. Units and weights colored in red denote the ones that are retrained, and black ones are ones that remain fixed. (b) Non-retraining models such as Progressive Network (Rusu et al., 2016) expands the network for the new task $t$, while withholding modification of network weights for previous tasks. (c) Our DEN selectively retrains the old network, expanding its capacity when necessary, and thus dynamically deciding its optimal capacity as it trains on.

parameters (Rusu et al., 2016). Our strategy is different from both approaches, since we retrain the network at each task $t$ such that each new task utilizes and changes only the relevant part of the previous trained network, while still allowing to expand the network capacity when necessary. In this way, each task $t$ will use a different subnetwork from the previous tasks, while still sharing a considerable part of the subnetwork with them. Figure 1 illustrates our model in comparison with existing deep lifelong learning methods.

There are a number of challenges that need to be tackled for such incremental deep learning setting with selective parameter sharing and dynamic layer expansion.

1) Achieving scalability and efficiency in training: If the network grows in capacity, training cost per task will increasingly grow as well, since the later tasks will establish connections to a much larger network. Thus, we need a way to keep the computational overhead of retraining to be low.

2) Deciding when to expand the network, and how many neurons to add: The network might not need to expand its size, if the old network sufficiently explains the new task. On the other hand, it might need to add in many neurons if the task is very different from the existing ones. Hence, the model needs to dynamically add in only the necessary number of neurons.

3) Preventing *semantic drift*, or *catastrophic forgetting*, where the network drifts away from the initial configuration as it trains on, and thus shows degenerate performance for earlier examples/tasks. As our method retrains the network, even partially, to fit to later learned tasks, and add in new neurons which might also negatively affect the prior tasks by establishing connections to old subnetwork, we need a mechanism to prevent potential semantic drift.

To overcome such challenges, we propose a novel deep network model along with an efficient and effective incremental learning algorithm, which we name as Dynamically Expandable Networks (DEN). In a lifelong learning scenario, DEN maximally utilizes the network learned on all previous tasks to efficiently learn to predict for the new task, while dynamically increasing the network capacity by adding in or splitting/duplicating neurons when necessary. Our method is applicable to any generic deep networks, including convolutional networks.

We validate our incremental deep neural network for lifelong learning on multiple public datasets, on which it achieves similar or better performance than the model that trains a separate network for each task, while using only $11.9\%p - 60.3\%p$ of its parameters. Further, fine-tuning of the learned network on all tasks obtains even better performance, outperforming the batch model by as much as $0.05\%p - 4.8\%p$. Thus, our model can be also used for structure estimation to obtain optimal performance over network capacity even when batch training is possible, which is a more general setup.

## 2 RELATED WORK

**Lifelong learning**    Lifelong learning (Thrun, 1995) is the learning paradigm for continual learning where the model learns from a sequence of tasks while transferring knowledge obtained from earlier tasks to later ones. Since its inception of idea by  Thrun (1995), it has been extensively studied due to its practicality in scenarios where the data arrives in streams, such as in autonomous driving or

learning of robotic agents. Lifelong learning is often tackled as an online multi-task learning problem, where the focus is on efficient training as well as on knowledge transfer. Eaton & Ruvolo (2013) suggest an online lifelong learning framework (ELLA) that is based on an existing multi-task learning formulation (Kumar & Daume III, 2012) that efficiently updates latent parameter bases for a sequence of tasks, by removing dependency to previous tasks for the learning of each task predictor, and preventing retraining previous task predictors. Recently, lifelong learning is studied in deep learning frameworks; since lifelong learning of a deep network can be straightforwardly done by simple re-training, the primary focus of research is on overcoming catastrophic forgetting (Kirkpatrick et al., 2017; Rusu et al., 2016; Zenke et al., 2017; Lee et al., 2017).

**Preventing catastrophic forgetting**    Incremental or lifelong learning of deep networks results in the problem known as catastrophic forgetting, which describes the case where the retraining of the network for new tasks results in the network forgetting what are learned for previous tasks. One solution to this problem is to use a regularizer that prevents the new model from deviating too much from the previous one, such as $\ell_2$-regularizer. However, use of the simple $\ell_2$-regularizer prevents the model from learning new knowledge for the new tasks, which results in suboptimal performances on later tasks. To overcome this limitation, Kirkpatrick et al. (2017) proposed a method called Elastic Weight Consolidation (EWC) that regularizes the model parameter at each step with the model parameter at previous iteration via the Fisher information matrix for the current task, which enables to find a good solution for both tasks. Zenke et al. (2017) proposed a similar approach, but their approach computes the per-synapse consolidation online, and considers the entire learning trajectory rather than the final parameter value. Another way to prevent catastrophic forgetting is to completely block any modifications to the previous network, as done in Rusu et al. (2016), where at each learning stage the network is expanded with a subnetwork with fixed capacity that is trained with incoming weights from the original network, but without backpropagating to it.

**Dynamic network expansion**    There are few existing works that explored neural networks that can dynamically increase its capacity during training. Zhou et al. (2012) propose to incrementally train a denoising autoencoder by adding in new neurons for a group of difficult examples with high loss, and later merging them with other neurons to prevent redundancy. Recently, Philipp & Carbonell (2017) propose a nonparametric neural network model which not only learns to minimize the loss but also find the minimum dimensionality of each layer that can reduce the loss, with the assumption that each layer has infinite number of neurons. Cortes et al. (2016) also propose a network that can adaptively learn both the structure and the weights to minimize the given loss, based on boosting theory. However, none of these work considered multi-task setting and involves iterative process of adding in neurons (or sets of neurons), while our method only needs to train the network once for each task, to decide how many neurons to add. Xiao et al. (2014) propose a method to incrementally train a network for multi-class classification, where the network not only grows in capacity, but forms a hierarchical structure as new classes arrive at the model. The model, however, grows and branches only the topmost layers, while our method can increase the number of neurons at any layer.

## 3    INCREMENTAL LEARNING OF A DYNAMICALLY EXPANDABLE NETWORK

We consider the problem of incremental training of a deep neural network under the lifelong learning scenario, where unknown number of tasks with unknown distributions of training data arrive at the model in sequence. Specifically, our goal is to learn models for a sequence of $T$ tasks, $t = 1, \ldots, t, \ldots, T$ for *unbounded* $T$ where the task at time point $t$ comes with training data $\mathcal{D}_t = \{\boldsymbol{x}_i, y_i\}_{i=1}^{N_t}$. Note that each task $t$ can be either a single task, or comprised of set of subtasks. While our method is generic to any kinds of tasks, for simplification, we only consider the binary classification task, that is, $y \in \{0, 1\}$ for input feature $\boldsymbol{x} \in \mathbb{R}^d$. It is the main challenge in the lifelong learning setting that all the previous training datasets up to $t-1$ are not available at the current time $t$ (only the model parameters for the previous tasks are accessible, if any). The lifelong learning agent at time $t$ aims to learn the model parameter $\boldsymbol{W}^t$ by solving following problem:

$$\underset{\boldsymbol{W}^t}{\text{minimize}}\ \mathcal{L}(\boldsymbol{W}^t; \boldsymbol{W}^{t-1}, \mathcal{D}_t) + \lambda\Omega(\boldsymbol{W}^t), \quad t = 1, \ldots \quad (1)$$

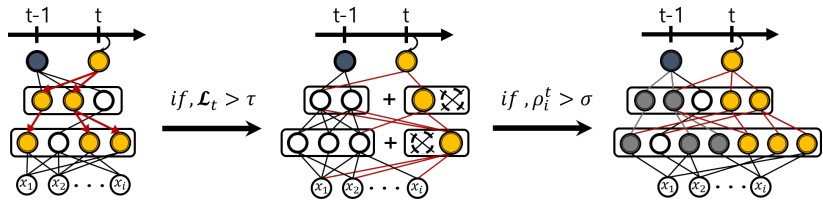

Figure 2: **Incremental learning of a dynamically expandable network: Left:** *Selective retraining.* DEN first identifies neurons that are relevant to the new tasks, and selectively retrains the network parameters associated with them. **Center:** *Dynamic network expansion.* If the selective retraining fails to obtain desired loss below set threshold, we expand the network capacity in a top-down manner, while eliminating any unnecessary neurons using group-sparsity regularization. **Right:** *Network split/duplication.* DEN calculates the drift $\rho_i^t$ for each unit to identify units that have drifted too much from their original values during training and duplicate them.

where $\mathcal{L}$ is task specific loss function, $\boldsymbol{W}^t$ is the parameter for task $t$, and $\Omega(\boldsymbol{W}^t)$ is the regularization (e.g. element-wise $\ell_2$ norm) to enforce our model $\boldsymbol{W}^t$ appropriately. In case of a neural network which is our primary interest, $\boldsymbol{W}^t = \{\boldsymbol{W}_l\}_{l=1}^L$ is the weight tensor.

To tackle these challenges of lifelong learning, we let the network to maximally utilize the knowledge obtained from the previous tasks, while allowing it to dynamically expand its capacity when the accumulated knowledge alone cannot sufficiently explain the new task. Figure 2 and Algorithm 1 describes our incremental learning process.

---

**Algorithm 1** Incremental Learning of a Dynamically Expandable Network
___

**Input:** Dataset $\mathcal{D} = (\mathcal{D}_1, \ldots, \mathcal{D}_T)$, Thresholds $\tau, \sigma$
**Output:** $\boldsymbol{W}^T$
**for** $t = 1, \ldots, T$ **do**
  **if** $t = 1$ **then**
    Train the network weights $\boldsymbol{W}^1$ using Eq. 2
  **else**
    $\boldsymbol{W}^t = SelectiveRetraining(\boldsymbol{W}^{t-1})$ {Selectively retrain the previous network using Algorithm 2 }
    **if** $\mathcal{L}_t > \tau$ **then**
      $\boldsymbol{W}^t = DynamicExpansion(\boldsymbol{W}^t)$ {Expand the network capacity using Algorithm 3}
    $\boldsymbol{W}^t = Split(\boldsymbol{W}^t)$ {Split and duplicate the units using Algorithm 4 }
___

In following subsections, we describe each component of our incremental learning algorithm in detail: 1) Selective retraining, 2) Dynamic network expansion, and 3) Network split/duplication.

**Selective Retraining.** A most naive way to train the model for a sequence of tasks would be retraining the entire model every time a new task arrives. However, such retraining will be very costly for a deep neural network. Thus, we suggest to perform selective retraining of the model, by retraining only the weights that are affected by the new task. Initially($t$=1), we train the network with $\ell_1$-regularization to promote sparsity in the weights, such that each neuron is connected to only few neurons in the layer below:

$$\underset{\boldsymbol{W}^{t=1}}{\text{minimize}}\ \mathcal{L}(\boldsymbol{W}^{t=1}; \mathcal{D}_t) + \mu \sum_{l=1}^{L} \|\boldsymbol{W}_l^{t=1}\|_1 \qquad (2)$$

where $1 \leq l \leq L$ denotes the $l_{th}$ layer of the network, $\boldsymbol{W}_l^t$ is the network parameter at layer $l$, and $\mu$ is the regularization parameter of the element-wise $\ell_1$ norm for sparsity on $\boldsymbol{W}$. For convolutional layers, we apply $(2, 1)$-norm on the filters, to select only few filters from the previous layer.

Throughout our incremental learning procedure, we maintain $\boldsymbol{W}^{t-1}$ to be sparse, thus we can drastically reduce the computation overheads if we can focus on the subnetwork connected new task. To this end, when a new task $t$ arrives at the model, we first fit a sparse linear model to predict task $t$

using topmost hidden units of the neural network via solving the following problem:

$$\underset{\boldsymbol{W}_{L,t}^t}{\text{minimize}}\ \mathcal{L}(\boldsymbol{W}_{L,t}^t\,;\boldsymbol{W}_{1:L-1}^{t-1},\mathcal{D}_t) + \mu\|\boldsymbol{W}_{L,t}^t\|_1 \tag{3}$$

where $\boldsymbol{W}_{1:L-1}^{t-1}$ denotes the set of all other parameters except $\boldsymbol{W}_{L,t}^t$. That is, we solve this optimization to obtain the connections between output unit $o_t$ and the hidden units at layer $L$-1 (fixing all other parameters up to layer $L$-1 as $\boldsymbol{W}^{t-1}$). Once we build the sparse connection at this layer, we can identify all units and weights in the network that are affected by the training, while leaving the part of the network that are not connected to $o_t$ unchanged. Specifically, we perform breadth-first search on the network starting from those selected nodes, to identify all units (and input feature) that have paths to $o_t$. Then, we train only the weights of the selected subnetwork $S$, denoted as $\boldsymbol{W}_S^t$:

$$\underset{\boldsymbol{W}_S^t}{\text{minimize}}\ \mathcal{L}(\boldsymbol{W}_S^t\,;\boldsymbol{W}_{S^c}^{t-1},\mathcal{D}_t) + \mu\|\boldsymbol{W}_S^t\|_2 \tag{4}$$

We use the element-wise $\ell_2$ regularizer since the sparse connections have been already established[1]. This partial retraining will result in lower computational overhead and also help with avoiding negative transfer, since neurons that are not selected will not get affected by the retraining process. Algorithm 2 describes the selective retraining process.

---

**Algorithm 2** Selective Retraining

---

**Input:** Datatset $\mathcal{D}_t$, Previous parameter $\boldsymbol{W}^{t-1}$
**Output:** network parameter $\boldsymbol{W}^t$
Initialize $l \leftarrow L-1$, $S = \{o_t\}$
Solve Eq. 3 to obtain $\boldsymbol{W}_{L,t}^t$
Add neuron $i$ to $S$ if the weight between $i$ and $o_t$ in $\boldsymbol{W}_{L,t}^t$ is not zero.
**for** $l = L-1,\ldots,1$ **do**
    Add neuron $i$ to $S$ if there exists some neuron $j \in S$ such that $\boldsymbol{W}_{l,ij}^{t-1} \neq 0$.
Solve Eq. 4 to obtain $\boldsymbol{W}_S^t$

---

**Dynamic Network Expansion.**   In case where the new task is highly relevant to the old ones, or aggregated partial knowledge obtained from each task is sufficient to explain the new task, selective retraining alone will be sufficient for the new task. However, when the learned features cannot accurately represent the new task, additional neurons need to be introduced to the network, in order to account for the features that are necessary for the new task. Some existing work (Zhou et al., 2012; Rusu et al., 2016) are based on a similar idea. However, they are either inefficient due to iterative training that requires repeated forward pass (Zhou et al., 2012), or adds in constant number of units at each task $t$ without consideration of the task difficulty (Rusu et al., 2016) and thus are suboptimal in terms of performance and network capacity utility.

To overcome these limitations, we instead propose an efficient way of using group sparse regularization to dynamically decide how many neurons to add at which layer, for each task without repeated retraining of the network for each unit. Suppose that we expand the $l_{th}$ layer of a network with a constant number of units, say $k$, inducing two parameter matrices expansions: $\boldsymbol{W}_l^t = [\boldsymbol{W}_l^{t-1}; \boldsymbol{W}_l^{\mathcal{N}}]$ and $\boldsymbol{W}_{l-1}^t = [\boldsymbol{W}_{l-1}^{t-1}; \boldsymbol{W}_{l-1}^{\mathcal{N}}]$ for outgoing and incoming layers respectively, where $\boldsymbol{W}^{\mathcal{N}}$ is the expanded weight matrix involved with added neurons. Since we do not always want to add in all $k$ units (depending on the relatedness between the new task and the old tasks), we perform group sparsity regularization on the added parameters as follows:

$$\underset{\boldsymbol{W}_l^{\mathcal{N}}}{\text{minimize}}\ \mathcal{L}(\boldsymbol{W}_l^{\mathcal{N}}\,;\boldsymbol{W}_l^{t-1},\mathcal{D}_t) + \mu\|\boldsymbol{W}_l^{\mathcal{N}}\|_1 + \gamma\sum_g \|\boldsymbol{W}_{l,g}^{\mathcal{N}}\|_2 \tag{5}$$

where $g \in \mathcal{G}$ is a group defined on the incoming weights for each neuron. For convolutional layers, we defined each group as the activation map for each convolutional filter. This group sparsity regularization was used in Wen et al. (2016) and Alvarez & Salzmann (2016) to find the right number of neurons for a full network, while we apply it to the partial network. Algorithm 3 describes the details on how expansion works.

---

[1]We can add in $\ell_1$-norm here for further identification of necessary weights

After selective retraining is done, the network checks if the loss is below certain threshold. If not, then at each layer we expand its capacity by $k$ neurons and solve for Eq. 5. Due to group sparsity regularization in Eq. 5, hidden units (or convolutional filters) that are deemed unnecessary from the training will be dropped altogether. We expect that from this dynamic network expansion process, the model captures new features that were not previously represented by $\boldsymbol{W}_l^{t-1}$ to minimize residual errors, while maximizing the utilization of the network capacity by avoiding to add in too many units.

---

**Algorithm 3** Dynamic Network Expansion

---

**Input:** Datatset $\mathcal{D}_t$, Threshold $\tau$
Perform Algorithm 2 and compute $\mathcal{L}$
**if** $\mathcal{L} > \tau$ **then**
    Add $k$ units $\boldsymbol{h}^{\mathcal{N}}$ at all layers
    Solve for Eq. 5 at all layers
**for** $l = L-1, \ldots, 1$ **do**
    Remove useless units in $\boldsymbol{h}_l^{\mathcal{N}}$

---

**Network Split/Duplication.**    A crucial challenge in lifelong learning is the problem of *semantic drift*, or *catastrophic forgetting*, which describes the problem where the model gradually fits to the later learned tasks and thus forgets what it learned for earlier tasks, resulting in degenerate performance for them. The most popular yet simple way of preventing semantic drift is to regularize the parameters from deviating too much from its original values using $\ell_2$-regularization, as follows:

$$\underset{\boldsymbol{W}^t}{\text{minimize}}\ \mathcal{L}(\boldsymbol{W}^t; \mathcal{D}_t) + \lambda \|\boldsymbol{W}^t - \boldsymbol{W}^{t-1}\|_2^2 \tag{6}$$

where $t$ is the current task, and $\boldsymbol{W}^{t-1}$ is the weight tensor of the network trained for tasks $\{1, \ldots, t-1\}$, and $\lambda$ is the regularization parameter. This $\ell_2$ regularization will enforce the solution $\boldsymbol{W}^t$ to be found close to $\boldsymbol{W}^{t-1}$, by the degree given by $\lambda$; if $\lambda$ is small, then the network will be learned to reflect the new task more while forgetting about the old tasks, and if $\lambda$ is high, then $\boldsymbol{W}^t$ will try to preserve the knowledge learned at previous tasks as much as possible. Rather than placing simple $\ell_2$ regularization, it is also possible to weight each element with Fisher information (Kirkpatrick et al., 2017). Nonetheless, if number of tasks is large, or if the later tasks are semantically disparate from the previous tasks, it may become difficult to find a good solution for both previous and new tasks.

A better solution in such a case, is to *split* the neuron such that we have features that are optimal for two different tasks. After performing Eq. 6, we measure the amount of semantic drift for each hidden unit $i$, $\rho_i^t$, as the $\ell_2$-distance between the incoming weights at $t$-1 and at $t$. Then if $\rho_i^t > \sigma$, we consider that the meaning of the feature have significantly changed during training, and split this neuron $i$ into two copies (properly introducing new edges from and to duplicate). This operation can be performed for all hidden units in parallel. After this duplication of the neurons, the network needs to train the weights again by solving Eq. 6 since *split* changes the overall structure. However, in practice this secondary training usually converges fast due to the reasonable parameter initialization from the initial training. Algorithm 4 describes the algorithm for *split* operation.

---

**Algorithm 4** Network Split/Duplication

---

**Input:** Weight $\boldsymbol{W}^{t-1}$, Threshold $\sigma$
Perform Eq. 6 to obtain $\widetilde{\boldsymbol{W}}^t$
**for** all hidden unit $i$ **do**
    $\rho_i^t = \|\boldsymbol{w}_i^t - \boldsymbol{w}_i^{t-1}\|_2$
    **if** $\rho_i^t > \sigma$ **then**
        Copy $i$ into $i'$ ($\boldsymbol{w}'$ introduction of edges for $i'$)
Perform Eq. 6 with the initialization of $\widetilde{\boldsymbol{W}}^t$ to obtain $\boldsymbol{W}^t$

---

**Timestamped Inference.**    In both the network expansion and network split procedures, we timestamp each newly added unit $j$ by setting $\{\boldsymbol{z}\}_j = t$ to record the training stage $t$ when it is added to the network, to further prevent semantic drift caused by the introduction of new hidden units. At inference time, each task will only use the parameters that were introduced up to stage $t$, to prevent the old tasks from using new hidden units added in the training process. This is a more flexible

strategy than fixing the weights learned up to each learning stage as in Rusu et al. (2016), since early tasks can still benefit from the learning at later tasks, via units that are further trained, but not split.

## 4 EXPERIMENT

**Baselines and our model.** **1) DNN-STL.** Base deep neural network, either feedforward or convolutional, trained for each task separately.

**2) DNN-MTL.** Base DNN trained for all tasks at once.

**3) DNN.** Base DNN. All incremental models use $\ell_2$-regularizations.

**3) DNN-L2.** Base DNN, where at each task $t$, $\boldsymbol{W}^t$ is initialized as $\boldsymbol{W}^{t-1}$ and continuously trained with $\ell_2$-regularization between $\boldsymbol{W}^t$ and $\boldsymbol{W}^{t-1}$.

**4) DNN-EWC.** Deep network trained with elastic weight consolidation (Kirkpatrick et al., 2017) for regularization.

**5) DNN-Progressive.** Our implementation of the progressive network (Rusu et al., 2016), whose network weights for each task remain fixed for the later arrived tasks.

**6) DEN.** Our dynamically expandable network.

**Base network settings.** **1) Feedforward networks:** We use a two-layer network with 312-128 neurons with ReLU activations. **2) Convolutional networks.** For experiments on the CIFAR-100 dataset, we use a modified version of AlexNet (Krizhevsky et al., 2012) that has five convolutional layers (64-128-256-256-128 depth with $5 \times 5$ filter size), and three fully-connected layers (384-192-100 neurons at each layer).

All models and algorithms are implemented using the Tensorflow (Abadi et al., 2016) library. We will release our codes upon acceptance of our paper, for reproduction of the results.

**Datasets.** **1) MNIST-Variation.** This dataset consists of $62,000$ images of handwritten digits from $0$ to $9$. Unlike MNIST, the handwritten digits are rotated to arbitrary angles and has noise in the background, which makes the prediction task more challenging. We use $1,000/200/5,000$ images for train/val/test split for each class. We form each task to be one-versus-rest binary classification.

**2) CIFAR-100.** This dataset consists of $60,000$ images of 100 generic object classes(Krizhevsky & Hinton, 2009). Each class has $500$ images for training and $100$ images for test. We used a CNN as the base network for the experiments on this dataset, to show that our method is applicable to a CNN. Further, we considered each task as a set of 10 subtasks, each of which is a binary classification task on each class.

**3) AWA (Animals with Attributes).** This dataset consists of $30,475$ images of $50$ animals (Lampert et al., 2009). For features, we use DECAF features provided with the dataset, whose dimensionality is reduced to $500$ by PCA. We use random splits of 30/30/30 images for training/validation/test.

### 4.1 QUANTITATIVE EVALUATION

We validate our models for both prediction accuracy and efficiency, where we measure the efficiency by network size at the end of training and training time. We first report the average per-task performance of baselines and our models in the top row of Figure 3. DNN-STL showed best performances on AWA and CIFAR-100 dataset; they are expected to perform well, since they are trained to be optimal for each task, while all other models are trained online which might cause semantic drift. When the number of tasks is small, MTL works the best from knowledge sharing via multi-task learning, but when the number of tasks is large, STL works better since it has larger learning capacity than MTL. Our model, DEN, performs almost the same as these batch models, and even outperforms them on MNIST-Variation dataset. Retraining models combined with regularization, such as L2 and EWC do not perform well, although the latter outperforms the former. This is expected as the two models cannot dynamically increase their capacity. Progressive network works better than the two, but it underperforms DEN on all datasets. The performance gap is most significant on AWA, as larger number of tasks ($T = 50$) may have made it more difficult to find the appropriate network

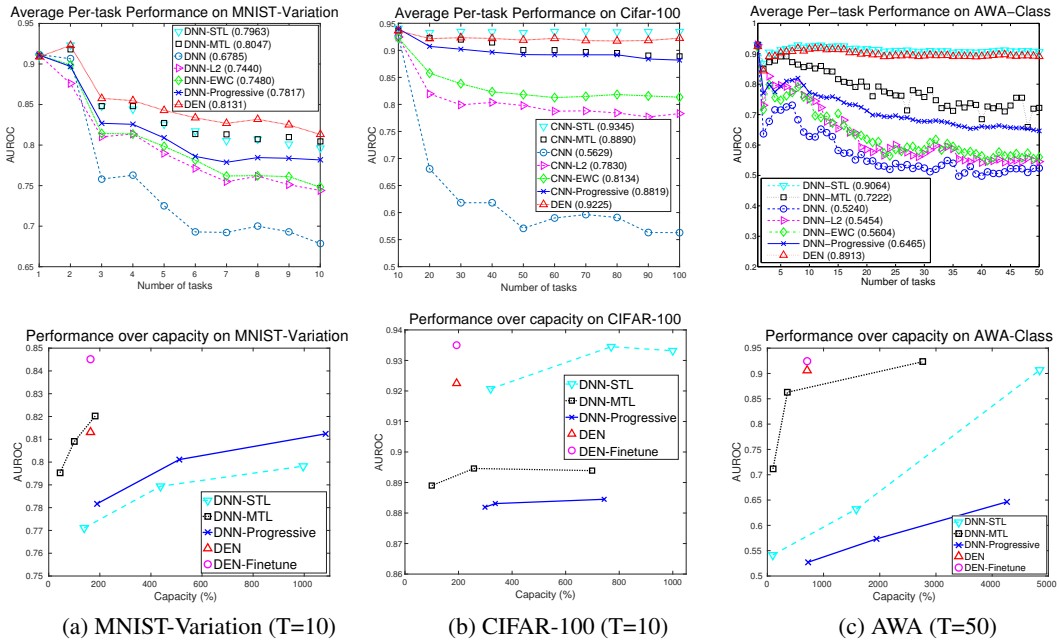

(a) MNIST-Variation (T=10)      (b) CIFAR-100 (T=10)      (c) AWA (T=50)

Figure 3: **Top row:** Average per-task performance of the models over number of task $t$, averaged over five random splits. The numbers in the legend denote average per-task performance after the model has finished learning $(t = T)$. **Bottom row:** Accuracy over network capacity. The network capacity is given relative to the capacity of MTL, which we consider as 100%.

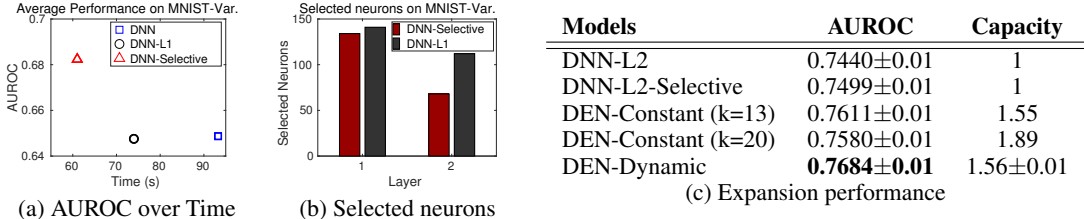

(a) AUROC over Time      (b) Selected neurons

| Models | AUROC | Capacity |
|---|---|---|
| DNN-L2 | 0.7440±0.01 | 1 |
| DNN-L2-Selective | 0.7499±0.01 | 1 |
| DEN-Constant (k=13) | 0.7611±0.01 | 1.55 |
| DEN-Constant (k=20) | 0.7580±0.01 | 1.89 |
| DEN-Dynamic | **0.7684±0.01** | 1.56±0.01 |

(c) Expansion performance

Figure 4: **Effect of selective retraining.** (a) shows AUROC over actual training time and (b) shows the number of selected neurons by selective retraining. **(c) Expansion performance.** We report both the prediction AUROC and network capacity measured by the relative number of parameters to that of DNN-MTL on MNIST-Variance dataset. Reported numbers are mean and standard error for five random splits.

capacity. If the network is too small, then it will not have sufficient learning capacity to represent new tasks, and if too large, it will become prone to overfitting.

We further report the performance of each model over network capacity measured relative to MTL on each dataset, in Figure 3 (bottom row). For baselines, we report the performance of multiple models with different network capacity. DEN obtains much better performance with substantially fewer number of parameters than Progressive network or obtain significantly better performance using similar number of parameters. DEN also obtains the same level of performance as STL using only 18.0%, 60.3%, and 11.9% of its capacity on MNIST-Variation, CIFAR-100, and AWA respectively. This also shows the main advantage of DEN, which is being able to dynamically find its optimal capacity, since it learns a very compact model on MNIST-Variation, whilst learning a substantially large network on CIFAR-100. Further fine-tuning of DEN on all tasks (DEN-Finetune) obtains the best performing model on all datasets, which shows that DEN is not only useful for lifelong learning, but can be also used for network capacity estimation when all tasks are available in the first place.

**Effect of selective retraining.** We further examine how efficient and effective the selective training is, by measuring the training speed and the area under ROC curve on MNIST-Variation dataset. To

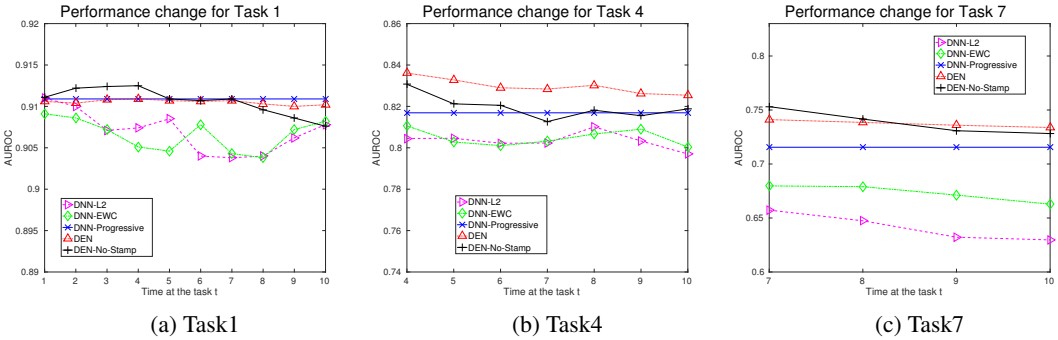

(a) Task1                    (b) Task4                    (c) Task7

Figure 5: **Semantic drift experiment on the MNIST-Variation dataset.** We report the AUROC of different models on $t = 1$, $t = 4$, and $t = 7$ at each training stage to see how the model performance changes over time for these tasks. Reported AUROC is the average over five random splits.

this end, we compare the model without network expansion, which we refer to as DNN-Selective, against retraining on DNN-L2 and DNN-L1 (Eq.(2)), for both the accuracy and efficiency. Figure 4(a) shows both the accuracy over training time measured as actual time spent with GPU computation, for each model. We observe that selective retraining takes significantly less time than the full retraining of the network, and even less than DNN-L1 that comes with sparse network weights. Further, whereas DNN-L1 obtained slightly less accuracy than DNN-L2, DNN-Selective improves the accuracy over the base network by $2\%p$. This accuracy gain may be due to the suppression of catastrophic forgetting, as DEN trains only a partial subnetwork for each newly introduced task. Figure 4(b) shows the number of selected neurons at each layer with selective retraining. Note that DNN-selective mostly selects less portion of upper level units which are more task-specific, while selecting larger portion of more generic lower layer units.

**Effect of network expansion.** We also compare the effectiveness of the network expansion against the model with a variant of our model that does selective retraining and layer expansion, but without network split. We refer to this model as DNN-Dynamic. We compare DNN-Dynamic with DNN-L2 used in the main experiment, and DNN-Constant, which is a version of our model that expands its capacity at each layer with fixed number of units, on MNIST-Variation dataset. Figure 4(c) shows the experimetal results. DNN-Dynamic obtains the best mean AU-ROC, significantly outperforming all models including DNN-Constant, while increasing the size of the network substantially less than DNN-Constant (k=20). This may be because having less number of parameters is not only beneficial in terms of training efficiency, but also advantageous in preventing the model from overfitting. We can set the network capacity of DNN-Constant to be similar (k=13) to obtain better accuracy, but it still underperforms DEN which can dynamically adjust the number of neurons at each layer.

**Effect of network split/duplication and timestamped inference.** To see how network split/duplication and unit timestamping help prevent semantic drift (or catastrophic forgetting), while allowing to obtain good performances on later tasks, we compare the performance of our model against baselines and also a variant of our DEN without timestamped inference (DEN-No-Stamp) at different learning stages. Each figure in Figure 5 (a), (b), and (c) shows how the performance of the model changes at each training stage $t$, for tasks $t=1$, $t=4$, and $t=7$.

We observe that DNN-L2 prevents semantic drift of the models learned at early stages, but results in increasingly worse performance on later tasks ($t=4, 7$). DNN-EWC, on the other hand, has better performance on later tasks than DNN-L2, as reported in Kirkpatrick et al. (2017). However, it shows significantly lower performance than both DNN-Progressive and our model, which may be due to its inability to increase network capacity, that may result in limited expressive power. DNN-Progressive shows no semantic drift on old tasks, which is expected because it does not retrain parameters for them. DEN w/o Timestamping works better than DNN-Progressive on later tasks, with slight performance degeneration over time. Finally, our full model with timestamped inference, DEN, shows no sign of noticeable performance degeneration at any learning stage, while significantly outperforming DNN-Progressive. This results show that DEN is highly effective in preventing semantic drift as well.

## 5 CONCLUSION

We proposed a novel deep neural network for lifelong learning, Dynamically Expandable Network (DEN). DEN performs partial retraining of the network trained on old tasks by exploiting task relatedness, while increasing its capacity when necessary to account for new knowledge required to account for new tasks, to find the optimal capacity for itself, while also effectively preventing semantic drift. We implement both feedforward and convolutional neural network version of our DEN, and validate them on multiple classification datasets under lifelong learning scenarios, on which they significantly outperform the existing lifelong learning methods, achieving almost the same performance as the network trained in batch while using as little as $11.9\%p - 60.3\%p$ of its capacity. Further fine-tuning of the models on all tasks results in obtaining models that outperform the batch models, which shows that DEN is useful for network structure estimation as well.

**Acknowledgements** This research was supported by Next-Generation Information Computing Development Program through the National Research Foundation of Korea of the Ministry of Science, ICT & Future Planning (NRF-2016M3C4A7952600), Samsung Research Funding Center of Samsung Electronics (SRFC-IT150203), and the ICT R&D program of MSIP/IITP (2016-0-00563, Research on Adaptive Machine Learning Technology Development for Intelligent Autonomous Digital Companion, and 2017-0-00537, Development of Autonomous IoT Collaboration Framework for Space Intelligence).

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

## APPENDIX A

### A.1 RESULTS ON PERMUTED MNIST

We provide additional experimental results on Permuted MNIST dataset Lecun et al. (1998). This dataset consists of $70,000$ images of handwritten digits from $0$ to $9$, where $60,000$ images are used for training, and $10,000$ images for test. The difference of this dataset from the original MNIST is that each of the ten tasks is the multi-class classification of a different random permutation of the input pixels. Figure 6 shows the results of this experiment. Our DEN outperforms all lifelong learning baselines while using only $1.39$ times of base network capacity. Further, DEN-Finetune achieves the best AUROC among all models, including DNN-STL and DNN-MTL.

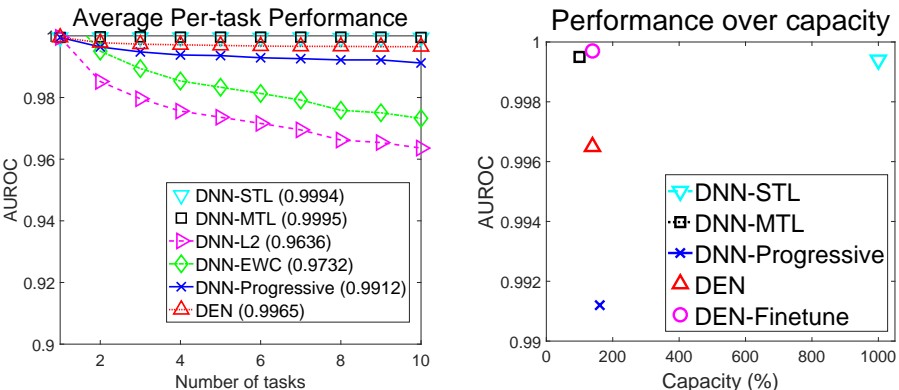

Figure 6: **Results on the Permuted MNIST.** Average per-task AUROC and network capacity of all models relative to MTL on Permuted MNIST.

