# OpenReview forum: "Lifelong Learning with Dynamically Expandable Networks"
_ICLR.cc/2018/Conference — Accept (Poster)_

### Official Review · AnonReviewer2 · 2017-11-27
**Interesting approach to continual learning**

**Rating:** 7
**Confidence:** 3

**Review:**

The paper was clearly written and pleasant to read. I liked the use of sparsity- and group-sparsity-promoting regularizers to select connections and decide how to expand the network.

A strength of the paper is that the proposed algorithm is interesting and intuitive, even if relatively complex, as it requires chaining a sequence of sub-algorithms. It was good to see the impact of each sub-algorithm studied separately (to some degree) in the experimental section. The results are overall strong.

It’s hard for me to judge the novelty of the approach though, as I’m not an expert on this topic.

Just a few points below:
- The experiments focus on a relevant continual learning problem, where each new task corresponds to learning a new class. In this setup, the method consistently outperforms EWC (e.g., Fig. 3), as well as the progressive network baseline.
Did the authors also check the performance on the permuted MNIST benchmark, as studied by Kirkpatrick et al. and Zenke et al.? It would be important to see how the method fares in this setting, where the tasks are the same, but the inputs have to be remapped, and network expansion is less of an issue.

- Fig. 4 would be clearer if the authors showed also the performance and how much the selected connection subsets would change if instead of using the last layer lasso + BFS, the full L1-penalized problem was solved, while keeping the rest of the pipeline intact.

- Still regarding the proposed selective retraining, the special role played by the last hidden layer seems slightly arbitrary. It may well be that it has the highest task-specificity, though this is not trivial to me. This special role might become problematic when dealing with deeper networks.

---

> ### Author Response · Authors · 2017-12-26
> **Answers to questions and the results on permuted MNIST dataset.**
>
> Q1. Did the authors also check the performance on the permuted MNIST benchmark, as studied by Kirkpatrick et al. and Zenke et al.? It would be important to see how the method fares in this setting, where the tasks are the same, but the inputs have to be remapped, and network expansion is less of an issue.
>
> A.  Following your suggestion, we have also experimented on the permuted MNIST dataset using feedforward network with 2 hidden layers and included the results in the Appendix section (See Figure 6). As expected, DEN achieves performance comparable to batch models such as STL or MTL, significantly outperforming both DNN-EWC and DNN-Progressive while obtaining a network that has significantly less number of parameters.
>
> Q2. Fig. 4 would be clearer if the authors showed also the performance and how much the selected connection subsets would change if instead of using the last layer lasso + BFS, the full L1-penalized problem was solved, while keeping the rest of the pipeline intact.
>
> A.   The suggested comparative study is already done in Fig. 4. DNN-L1 shows the results using the full L1-penalized regularizer instead of the last layer lasso + BFS. This result shows that selective retraining is indeed useful in reducing time complexity of training and perform selective knowledge transfer to obtain better accuracy.
>
>      For better understanding of the BFS process, we updated the figure that illustrates the selective retraining process to include arrows (Leftmost figure of Figure 2).
>
> Q3. Still regarding the proposed selective retraining, the special role played by the last hidden layer seems slightly arbitrary. It may well be that it has the highest task-specificity, though this is not trivial to me. This special role might become problematic when dealing with deeper networks.
>
> A. The last hidden layer is not the only layer that is learned to be task-specific, as the BFS process selects units that are useful for the given task at all layers of the network and retrains them.
>
>      To show that selective retraining does not become problematic with deeper networks, we performed additional experiments on the CIFAR-100 dataset with a 8-layer network which is a slight modification of AlexNet. The results show that DEN obtains similar performance gain over baselines even with this deeper network.

---

### Official Review · AnonReviewer1 · 2017-11-27
**Investigation of DENs and variants applied to lifelong learning**

**Rating:** 6
**Confidence:** 3

**Review:**

The topic is of great interest to the community, and the ideas explored by the authors are reasonable, but I found the conclusion less-than-clear. Mainly, I was not sure how to interpret the experimental findings, and did not have a clear picture of the various models being investigated (e.g. "base DNN regularized with l2"), or even of the criteria being examined. What is "learning capacity"? (If it's number of model parameters, the authors should just say, "number of parameters"). The relative performance of the different models examined, plotted in the top row of Figure 3, is quite different, and though the authors do devote a paragraph to interpreting the results, I found it slightly hard to follow, and was not sure what the bottom line was.

What does the "batch model" refer to?

re. " 11.9%p − 51.8%p"; remove "p"?

Reference for CIFAR-100? Explain abbreviation for both CIFAR-100 and AWA-Class?

re. "... but when the number of tasks is large, STL works better since it has larger learning capacity than MTL": isn't the number of parameters matched? If so, why is the "learning capacity" different? What do the authors mean exactly by "learning capacity"?

re. Figure 3, e.g. "Average per-task performance of the models over number of task t": this is a general point, but usually the expression "<f(x)> vs. <x>" is used rather than "<f(x)> over <x>" when describing a plot.

"DNN: dase (sic) DNN": how is this trained?

---

> ### Author Response · Authors · 2017-12-26
> **Answers to the questions**
>
> Q1. What does the "batch model" refer to?
> A.  “Batch model” refers to models that are not trained in an incremental manner; in other words, a batch model is trained with all tasks at hand, such as DNN-MTL or DNN-STL.
>
> Q2. re. " 11.9%p − 51.8%p"; remove "p"?
> A. %p stands for percent point and is a more accurate way of denoting absolute performance improvements compared to %.
>
> Q3. Reference for CIFAR-100? Explain abbreviation for both CIFAR-100 and AWA-Class?
> A.   Thank you for the suggestion. We updated the reference for the CIFAR-100 dataset and included the full dataset name for AWA in the revision. CIFAR is simply a dataset
>
>
> Q4. re. "... but when the number of tasks is large, STL works better since it has larger learning capacity than MTL": isn't the number of parameters matched? If so, why is the "learning capacity" different? What do the authors mean exactly by "learning capacity"?
> A.   By “learning capacity”, we are referring to the number of parameters in a network. DNN-MTL learns only a single network for all T tasks whereas DNN-STL learns T networks for T tasks. For the experiments that generated the plots in the top row of Figure 3, we used the same network size for both DNN-STL and DNN-MTL, and therefore, DNN-STL used T times more parameters than DNN-MTL. For accuracy / network capacity experiments in the bottom row, we diversified the base network capacity for both baselines.
>
> Q5. "DNN: dase (sic) DNN": how is this trained?
> A.   Thank you for pointing out the typo. We have corrected it in the revision.

---

### Official Review · AnonReviewer3 · 2017-11-30
**Method for lifelong learning with neural networks**

**Rating:** 8
**Confidence:** 2

**Review:**

In this paper, the authors propose a method (Dynamically Expandable Network) that addresses issues of training efficiency, how to dynamically grow the network, and how to prevent catastrophic forgetting.

The paper is well written with a clear problem statement and description of the method for preventing each of the described issues. Interesting points include the use of an L1 regularization term to enforce sparsity in the weights, as well as the method for identifying which neurons have “drifted” too far and should be split. The use of timestamps is a clever addition as well.

One question would be how sparse training is done, and how this saves computation, especially with the breadth-first search described on page 5. A critique would be that the base networks (a two layer FF net and LeNet) are not very compelling.

Experiments indicate that the method works well, with a clear improvement over progressive networks. Thus, though there isn’t one particular facet of the paper that leaps out, overall the method and results seem solid and worthy of publication.

---

> ### Author Response · Authors · 2017-12-26
> **Description of breath-first search and the experimental results with deeper networks**
>
> Q1. One question would be how sparse training is done, and how this saves computation, especially with the breadth-first search described on page 5.
>
> A. Sparse training is done at both initial network training (Eq. (2)) and selective retraining step (Eq. (3)), using L1 regularizer. First, the initial network training obtains a network with sparse connectivity between neurons at consecutive layers.
>
> Then, the selective retraining selects the neurons at the layer just before the output neurons of this sparse network, and then using the topmost layer neurons as starting vertices, it selects neurons at each layer that have connections to the selected upper-layer neurons (See the leftmost model illustration of Figure 2).
>
> This results in obtaining a subnetwork of the original network that has much less number of parameters (Figure 4.(b)) that can be trained with significantly less training time (Figure 4.(a)). The selected subnetwork also obtains substantially higher accuracy since it leverages only the relevant parts of the network for the given task (Figure 4.(a)).
>
> Q2. A critique would be that the base networks (a two layer FF net and LeNet) are not very compelling.
>
> A. To show that our algorithm obtains performance improvements on any generic networks, we experimented with a larger network that consists of 8 layers, which is a slight modification of AlexNet. With this larger network, our algorithm achieved similar performance gain over the baseline models as in the original experiments. We included the new results in the revision.

---

### Author Response · Authors · 2018-01-05
**Summary of the updates**

We summarize the updates made in the revision below:

Main updates:
- To address the comment from AnonReviewer 3 that the base networks are too small, we replaced the experimental results on CIFAR-100 dataset with results from a deeper network, that is a slight modification of AlexNet (8 layers: 5 Conv and 3 FC). Our algorithm achieved significant performance gain over the baseline models on this dataset as well.

- Following the suggestion from AnonReviewer 2, we performed new experiments on the Permuted MNIST dataset with feedforward networks, and included the experimental results in the Appendix Section. The results show DEN achieves comparable performance to batch models (STL and MTL), while significantly outperforming both DNN-EWC and DNN-Progressive.

Page 2: Updated the value of %p to reflect the updates in the CIFAR-100 experimental results.
Page 7: 1) Corrected the typo: 3) DNN. Dase → 3) DNN. Base
              2) Replaced the description of LeNet with the description of the deeper CNN used in the new experiments.
Page 8: Updated the CIFAR-100 plot in Figure 3 with the results from the modified AlexNet.
Pages 7-8, 10: Updated the value of %p to reflect the updates in the CIFAR-100 results.
Page 11: Included the Appendix section, which contains the results and discussions of the Permuted MNIST experiments.

---

### Decision · Program_Chairs · 2018-01-29
**ICLR 2018 Conference Acceptance Decision**

**Decision:**

Accept (Poster)

**Comment:**

PROS:
1. good results; the authors made it work
2. paper is largely well written

CONS:
1. some found the writing to be unclear and sloppy in places
2. the algorithm is complicated -- a chain of sub-algorithms

A few small points:

-I initially found Algorithm 1 to be confusing because it wasn't clear whether it was intended to be invoked for each task (making the training depend on all the datasets).  I finally convinced myself that this was not the intention and that the inner loop of the algorithm is what is actually executed incrementally.